# Therapeutic Application of Extracellular Vesicles-Capsulated Adeno-Associated Virus Vector via *nSMase2/Smpd3*, Satellite, and Immune Cells in Duchenne Muscular Dystrophy

**DOI:** 10.3390/ijms23031551

**Published:** 2022-01-28

**Authors:** Yasunari Matsuzaka, Yukihiko Hirai, Kazuo Hashido, Takashi Okada

**Affiliations:** 1Division of Molecular and Medical Genetics, Center for Gene and Cell Therapy, The Institute of Medical Science, University of Tokyo, Minato-ku, Tokyo 108-8639, Japan; hirai@ims.u-tokyo.ac.jp; 2Administrative Section of Radiation Protection, National Institute of Neuroscience, National Center of Neurology and Psychiatry, Kodaira, Tokyo 187-8551, Japan; hashido@ncnp.go.jp

**Keywords:** adeno-associated virus, Duchenne muscular dystrophy, extracellular vesicle, extracellular vesicle-capsulated adeno-associated virus vector, microRNAs, myofiber degeneration, *nSMase2/Smpd3*, satellite cell

## Abstract

Duchenne muscular dystrophy (DMD) is caused by loss-of-function mutations in the dystrophin gene on chromosome Xp21. Disruption of the dystrophin–glycoprotein complex (DGC) on the cell membrane causes cytosolic Ca^2+^ influx, resulting in protease activation, mitochondrial dysfunction, and progressive myofiber degeneration, leading to muscle wasting and fragility. In addition to the function of dystrophin in the structural integrity of myofibers, a novel function of asymmetric cell division in muscular stem cells (satellite cells) has been reported. Therefore, it has been suggested that myofiber instability may not be the only cause of dystrophic degeneration, but rather that the phenotype might be caused by multiple factors, including stem cell and myofiber functions. Furthermore, it has been focused functional regulation of satellite cells by intracellular communication of extracellular vesicles (EVs) in DMD pathology. Recently, a novel molecular mechanism of DMD pathogenesis—circulating RNA molecules—has been revealed through the study of target pathways modulated by the Neutral sphingomyelinase2/Neutral sphingomyelinase3 (*nSMase2/Smpd3*) protein. In addition, adeno-associated virus (AAV) has been clinically applied for DMD therapy owing to the safety and long-term expression of transduction genes. Furthermore, the EV-capsulated AAV vector (EV-AAV) has been shown to be a useful tool for the intervention of DMD, because of the high efficacy of the transgene and avoidance of neutralizing antibodies. Thus, we review application of AAV and EV-AAV vectors for DMD as novel therapeutic strategy.

## 1. Introduction

Duchenne muscular dystrophy (DMD) is a severe and progressive muscular disorder, which mainly manifests as degeneration and regeneration in skeletal and cardiac muscles and, finally, leads to myofiber necrosis, fibrosis, and muscle weakness, through membrane fragility and disrupted cell signaling [1,2,3,4,5]. The incidence of DMD is estimated to be approximately 1 in 3500 to 10,000 newborn and adolescent males, with a prevalence of less than 10 cases per 100,000 males, which does not differ between regions [5,6,7,8,9,10,11]. Typically, the clinical symptoms of this disease are first recognized by muscular weakness, with difficulties in climbing stairs, a waddling gait, and frequent falls under five years of age, subsequently progressing into the loss of ability of independent ambulation, with wheelchair dependency around 10 years of age, eventually leading to premature death, due to respiratory and cardiac failure at approximately 20–40 years of age [5,12,13,14,15].

In DMD patients, sensitive damage and degeneration of muscle fibers, through sarcolemma destabilization and progressive loss of muscular repair ability via stem cells, concomitantly increase inflammation and fibrosis in muscular tissues, due to loss-of-function with frameshifting or nonsense mutations in the dystrophin gene on chromosome Xp21, which codes for the protein dystrophin, a myofiber membrane protein [1,16,17,18,19,20,21,22,23]. In a study of thousands of different mutations in DMD patients, the categories of each mutation were deletions, duplications, and point mutations, including small deletions and insertions, comprise of approximately 60–70%, 5–15%, and 20%, respectively [5,24,25]. In addition, two hotspot regions of these deletion and duplication clusters in the DMD patients are located at exons 45–55 (approximately 47%) and exons 3–9 (approximately 7%), respectively [5,26,27]. In addition, de novo germline mutations were observed in one-third of DMD patients; in these cases, the mothers were not somatic carriers of DMD mutations but had children with DMD [5,28,29]. The frequency of germline mutations in oocytes and sperms varies per individual but appears to be up to 14% [5,30].

The full functional dystrophin protein forms a complex known as the dystrophin-associated protein complexes (DAPCs), with ten kinds of proteins, including the dystroglycan subcomplex (α-dystroglycan and β-dystroglycan), the sarcoglycan subcomplex (α-sarcoglycan, β-sarcoglycan, γ-sarcoglycan, and δ-sarcoglycan), sarcospan, syntrophin, dystrobrevin, and neuronal nitric oxide synthase (nNOS) [5,31,32,33,34,35,36,37,38]. These differences are dependent not only on the type of tissues or cell but also on the different regions of the same myocyte on the cell membrane. It is critical for maintaining structural stability and integrity of the muscle sarcolemma by connecting the internal cytoskeletal F-actin of a muscle fiber to the surrounding extracellular matrix via its N- and C-terminal domains and transmission of forces within the muscle [5,31,32,33,34,35,36,37,38]. In DMD, DAPC disassembly is caused by dystrophin deficiency, resulting in the loss of the interaction between F-actin and the extracellular matrix, and, finally, wide-ranging dysfunction in muscle cells, resulting in high susceptibility to contraction damage [5]. The mislocation of nNOS from the sarcolemma to the sarcoplasm leads to a reduction in the total cellular level of nNOS, which decreases the production and secretion of nitric oxide to the surrounding vasculature and results in ischemic damage to muscle [5,39]. This mechanism, in which loss of sarcolemmal nNOS increases intracellular calcium concentrations via Ryanodibe receptor 1 (RyR1) leakage or nicotinamide adenine dinucleotide phosphate oxidases (NOX)-derived cytosolic reactive oxygen species (ROS) production, is supported by sarcolemmal tears (delta lesions) and rupture, which can be detected by passive leakage of muscle enzymes and microRNAs from muscles into the bloodstream [32,39,40,41].

The myofiber-specific genetic ablation of dystroglycan in mice does not result in dystrophy-like muscle degeneration; however, mice with muscle stem cells (also called satellite cells, SCs) exhibit specific loss of dystroglycan, with markedly delayed muscle regeneration [42]. In addition, SCs lacking dystrophin markedly increase the number of abnormal nonpolarized mitotic divisions and reduced asymmetric cell divisions, exacerbating dystrophic pathology [43,44]. These reports suggest that SC dysfunction is mainly involved in muscular dystrophy. In addition, it has been suggested that myofiber instability, leading to disruption of the dystrophin protein lining inside the cell membrane, is not the only cause of dystrophic degeneration but rather that the phenotype might be caused by multiple factors, including stem cell and myofiber functions. Thus, we review novel therapeutic strategy for DMD.

## 2. Therapy Strategy of DMD

To date, there are mainly two of treatment strategies for DMD have been explored: (1) the treatment for actual cause of DMD–dystrophin deficiency (dystriphin-targeted therapies), such as adeno-associated virus (AAV)-mediated micro/minidystrophin gene delivery, synthetic antisense oligonucleotides for exon skipping, nonsense readthrough, CRISPR-Cas9 (clustered regularly interspaced short palindromic repeat-CRISPR-associated protein 9)-mediated genome editing, protein replacement therapies, and primarily utropin; (2) the therapy of downstream pathological changes, such as transplantation of muscular stem cells, corticosteroids (prednisolone or deflazacort) with highly effective in therapy of concomitant destructive processes (namely inflammation) and an improvement in calcium homeostasis, leading to a decrease in oxidative stress in muscle tissue, and in mitochondrial function and biogenesis [45,46,47,48,49,50,51,52,53,54,55] (Figure 1). Currently, gene editing is attracting attention as a therapeutic strategy for DMD because of the restoration of the dystrophin reading frame in more than 40% of all patients with DMD [56,57,58,59,60,61,62,63]. However, it has been reported that clinical trials with this strategy have to be discontinued because of off-target effects that cause chromosomal abnormalities. In addition, the desired mutation may not be introduced in all cells, resulting in a somatic mosaic mutation, where it is difficult to undo edited parts of the genome. Additionally, as no trace of genome editing remains, it is difficult to investigate the cause of an adverse event. Therefore, the development of an additive gene therapy, as a safe protein supplementation, should be prioritized.

AAV vectors are derived from non-pathogenic viruses and capable of transferring genes into non-dividing cells. Moreover, because gene expression in target cells is sustainable for a long period of time, it is also excellent, in terms of safety and efficacy. Therefore, it is attracting attention as a clinically applied vector for gene therapy in DMD. Dysfunctions in skeletal and cardiac muscles of DMD model *mdx* mice are significantly attenuated by partial restoration of dystrophin protein using the micro-dystrophin protein, which delivers a part of the cDNA copy of dystrophin to the affected tissues, based on internally deleted dystrophins [64,65,66,67,68,69,70,71,72,73,74,75,76,77,78,79]. However, there are some problems with the administration of AAV vectors into the skeletal muscle of canine X-linked muscular dystrophy model in Japan (CXMDj), which shows severe dystrophic phenotypes without immunosuppression, resulting in insufficient transgene expression with potent innate immune responses [80,81]. Furthermore, the AAV vector has a limited carrying capacity of ~4.7 kb or less, because it is a vector derived from a small virus, whereas the muscle isoform of dystrophin, Dp427, is encoded by ~11.4 kb cDNA [82]. The time to reach the gene expression peak was long because of the single-stranded DNA genome. The gene transfer efficiency was reduced by the neutralizing antibody against the viral capsid of the AAV vector. The cost for large-scale production of AAV vectors is expensive, and large-scale protocols have not yet been established. In addition, deaths, carcinogenicity, hepatotoxicity, neurotoxicity, and thrombotic microangiopathy have been reported in clinical trials with high doses of AAV vectors, including micro-dystrophin [82,83,84,85].

In gene therapy using an exon skipping agent, the low efficiency of introduction into cardiomyocytes has become an issue [86,87]. In addition, it is impossible to treat all mutations in the dystrophin gene at present because the drug corresponds only to mutations in a limited exon region [88,89,90]. Besides, corticosteroid treatment of patients with DMD commonly showed adverse effects, such as a higher risk of developing respiratory and circulatory dysfunctions earlier, rather than later [91]. Thus, there is currently no complete and radical cure, and the establishment of new therapies is eagerly awaited.

## 3. Satellite and Immune Cells

A single intravenous dose of cardiac stromal cells, termed cardiosphere-derived cells (CDCs), which are cardiac progenitor/stromal cells with anti-inflammatory, antioxidant, antifibrotic, and cardiomyogenic properties [92,93], directly into the muscles of *mdx* mice improves the dystrophic phenotype via augmentation of cardiac and skeletal muscles [94]. Furthermore, extracellular vesicles (EVs), which are membrane microvesicles, approximately 30–100 nm in size, that are generated from multivesicular bodies of the terminal endosomal pathway, recapitulate the therapeutic benefits of the CDCs because the blockage of EV biogenesis fails to improve cardiac and skeletal muscles in vivo [93]. Both CDC and EV treatment in *mdx* partially reverse heart damage by attenuating fibrosis, as well as by decreasing nuclear factor-kappa B, NF-κB phosphorylation, and macrophage infiltration. Moreover, these treatments enhance cardiomyogenesis, normalize mitochondrial protein deficits, and improve the pathology of skeletal muscles by increasing force and promoting skeletal myogenesis with the activation of SCs, which proliferate and differentiate into myoblasts to form myotubes with reduced interstitial fibrosis [94].

Furthermore, adult skeletal muscles have a self-repair capacity, modulated by SCs and immune cells [95]. Macrophages modify myoblast proliferation and their commitment to differentiated myocytes, as well as the formation of mature myotubes. This is done by enhancing muscle growth through the interaction of differentially activated myogenic precursor cells (MPCs) with anti-inflammatory macrophages that may exert deleterious effects, due to the resolution of damaged tissues by removal of necrotic cells [89,90,91,92]. In general, macrophage infiltration in acutely injured muscle must terminate clearing of its damaged tissues, so as to prevent the destruction of the adjacent undamaged tissues. However, in chronic inflammatory conditions of DMD, macrophages persistently accumulate and contribute to disease progression by disturbing myogenesis, with SCs coordinated by macrophage polarization [94,96,97,98,99,100,101,102,103,104,105,106,107,108,109]. In particular, loss of the Kruppel-like factor 2 *(KLF2*) gene, in myeloid-derived cells, enhances the inflammatory immune response to muscle injury through the recruitment of greater numbers of inflammatory Ly6C-positive monocytes, which provide monocyte-derived mature Ly6C-, CD11b-, and F4/80-positive macrophages in circulation and enable regeneration via the activation of SCs and myogenesis [103]. During muscle regeneration, the normal transition from the pro- to anti-inflammatory phase, which is mainly controlled by macrophages, is required for spatiotemporal regulation of SC differentiation, angiogenesis, and matrix remodeling [104]. The initiation of the inflammatory response induces extravasation of leukocytes, of which Ly6C-positive, Ccr2-positive, and Cx3cr1-low circulating monocytes in the bloodstream cross the endothelium because of increased vascular permeability. Subsequently, the monocytes enter the damaged muscular tissues, and neutrophils mount the proinflammatory response to attract Ly6C-positive monocytes, mainly through the Ccl2-Ccr2 axis, thus clearing the surrounding debris via efferocytosis and removing the apoptotic cells via phagocytic cells [110,111,112,113,114]. In contrast, Ly6C-positive, Ccr2-positive, and Cx3cr1-high monocytes patrol to survey vessel wall integrity [110,113]. After the debris of muscular tissue is cleansed by phagocytosis, proliferated macrophages exhibit an anti-inflammatory/restorative response, which is characterized by myogenesis, growth of new myofibers, angiogenesis, and matrix remodeling, through interactions of the macrophages with surrounding cells, as in accordance with the shift of Ly6C-positive into Ly6C-negative within the damaged tissue [98,106,115,116,117,118,119,120]. The shift in macrophage status from pro- to anti-inflammatory was regulated by mediators of the resolution of inflammation, such as annexin A1 (AnxA1), developmental endothelial locus-1 (DEL-1), glucocorticoid-induced leucine zipper (GILZ), and secretory leukocyte peptidase inhibitor (SLPI), by which recognition and engulfment of dead cells triggers the transcription of anti-inflammatory effectors, such as transforming growth factor-β (TGF-β), interleukin (IL)-10, and specialized pro-resolving mediators, concomitant with the downregulation of inflammatory cytokines [110,121,122].

Furthermore, the resolution of inflammation during skeletal muscle regeneration in vivo is controlled by several molecular pathways that regulate macrophage conversion, including AMP-activated protein kinase, alpha 1 (AMPKα1), BTB and CNC homology 1, basic leucine zipper transcription factor 1 (Bach1), insulin growth factor-1 (IGF-1), mitogen-activated protein kinase phosphatase 1/dual specificity phosphatase 1 (Mkp-1/Dusp1), peroxisome proliferator activated receptor gamma (Ppar-γ), and scavenger receptor class B, member 1 (SRBI) [99,108,109,116,123,124,125]. Among them, loss of the *Bach1* gene in myeloiod cells enhances the muscular regeneration process during acute muscular damage through the conversion of inflammatory Ly-6C-high and F4/80-high marcorphages to repair Ly-6C-low and F4/80-high macrophages. This conversion is mediated by the induced expression of heme oxygenase 1 (HO-1) and regulation of many key inflammatory and repair-related genes, such as IGF-1, solute carrier family 40 member 1 (Slc40a1), IL-6, IL-10, growth differentiation factor 3 (Gdf3), Ppar-γ, Dusp1, and CCAAT/enhancer binding protein (C/EBP), beta (Cebpb) [117]. In addition, phagocytosis of muscle debris participates in macrophage skewing: M1 macrophages are decreased upon the phagocytosis of necrotic and apoptotic muscular precursor cells, with decreased tumor necrosis factor-α (TNF-α) secretion, whereas M2 macrophages increased, in accordance with TGF-β secretion [99]. In contrast, the phagocytosis of macrophages in loss-of-function, myeloid-specific AMPKα1 was impaired, leading to a defect in their phenotypic transition into M2 macrophages, in which the impaired fusion ability and abnormal proliferation of myogenic precursor cells by M2a/c macrophages finally resulted in a delay of skeletal muscle regeneration [109]. However, these deficiencies were partially rescued by the transplantation of the bone marrow of the wild-type variant [109]. In addition, monocyte/macrophage-derived IGF-I coordinates macrophage polarization and myogenesis during muscle regeneration [94,108]. Moreover, Pparγ in macrophages regulates skeletal muscle regeneration via myoblast proliferation and differentiation by transactivation and secretion of Gdf3 [116]. However, the role of Pparγ in M2 polarization remains controversial. On the one hand, Pparγ activation in adipose tissues promotes M2 polarization [126]. On the other hand, the activation of Pparγ suppresses M2 polarization via the inactivation of cytotoxic T lymphocytes [127]. In addition, the conditional deletion of the *Cebpb* gene in muscle fibers shows normal regeneration, but the loss of the *Cebpb* gene in macrophages leads to defective M2 poralization, increased inflammation, and reduced ability of muscular regeneration [99].

Under hypoxic culture conditions, CDCs secrete EVs encapsulated with a high abundance of a microRNA, termed miR-148, which promotes the regeneration effects of EVs through differentiation of myogenic precursor [94]. The recruited Ly6C-positive monocytes/macrophages with pro-inflammatory profiles activate SCs by disrupting the quiescence of their niche during the expansion phase, leading to the formation of new functional myofibers, owing to the commitment of many progenitor cells to undergo myogenic differentiation and self-renewal of a subset to return to quiescence [101,121,128,129].

## 4. Non-Invasive Biomarkers

Non-invasive biomarkers that evaluate the progress of DMD pathology, following therapeutic interventions, are limited. Among potential biomarkers, such as proteins, nucleic acids, metabolites, polymorphisms, mutations, RNA splicing, and epigenetics, microRNAs (miRNAs), have been intensively studied. The RNA comprises of a class of small non-coding RNA molecules, 22 nucleotides in length, that can regulate gene expression at the post-transcriptional level by destabilizing mRNA and translation silencing. They have been selected because of their: (1) high content in body fluids, including serum, plasma, tear, lymph, breast milk, urine, semen, saliva, and sweat; (2) high-stability in the blood and outside the body, due to capsulation into EVs and formation of complex with RNA-binding proteins; (3) unique expression profile, correlated with pathology progression for monitoring the initial stage of the pathological condition, such as immediately before or after the onset; (4) high-sensitivity and specificity to suppress false positives and negatives; and (5) high-throughput and cost-reduction for evaluation systems of disease levels [130,131]. As for miRNAs in DMD as biomarkers, the levels of three myomiRs, namely miR-1, miR-133a, and miR-206, which are abundant miRNAs in muscles, were increased in the sera of patients and animal models of muscular dystrophy with restored levels, due to expression of functional dystrophin protein, which was shown to be inversely correlated with disease severity in DMD patients [132,133,134,135,136]. In addition, miR-1 and miR-133 are expressed in cardiac and skeletal muscles and involved in the proliferation and differentiation processes [137,138,139,140,141]. Moreover, mir-206 is a skeletal muscle-specific miRNA, expressed in SCs, involved in muscle development and regeneration [142]. In addition, in the muscle of *mdx* mice, miR-1 and miR-133a levels were shown to be downregulated, whereas miR-206 level was upregulated, recovering to wild-type levels by restoration of the dystrophin protein [133,134,135,136]. 

## 5. Neutral Sphingomyelinase 2/Sphingomyelin Phosphodiesterase 3 (*nSMase2/Smpd3*) and DMD

DMD exhibits inflammatory responses as a common feature, during which, after muscle injury, inflammatory macrophages are recruited to secrete inflammatory cytokines and miRNAs, due to repair and regeneration [143]. In particular, these myomiRs, encapsulated within EVs, which are produced by the biogenesis of ceramide from sphingomyelin, are released from cells into circulation and controlled by the *nSMase2/Smpd3*-regulated secretory machinery of EVs [144,145,146,147,148,149]. Thus, to elucidate the relationship between the release of myomiRs via EV and DMD pathogenesis, GW4869 (an inhibitor of *nSMase2/Smpd3*) was administered to *mdx* mice. It has been shown that the inhibition of *nSMase2/Smpd3* enzymatic activity ameliorates skeletal muscles of muscular dystrophy in *mdx* mice, in turn, inhibits ceramide synthesis [150]. However, there are problems with the GW4869 inhibitor, in that it has effects on other *nSMase2/Smpd3* family members and relatively short-term inhibitory effects. Therefore, to investigate the effects of *nSMase2/Smpd3* on dystrophic pathology, we generated *mdx* mice lacking the *nSMase2/Smpd3* gene (*mdx:Smpd3* double knockout [DKO] mice) [151]. Deletion of the *nSMase2/Smpd3* gene in *mdx* mice reduces inflammation in dystrophic muscles, as indicated by a reduction in the infiltration of excess inflammatory cells and decrease in inflammatory cytokine expression levels, such as TNF-α, CD68, CD45, chemokine (C-C motif) receptor 5 (Ccr5), IL-1 receptor antagonist (IL-1ra), and IL-6 [151]. In addition, disruption of the *nSMase2/Smpd3* gene attenuated muscle membrane permeability in dystrophic *mdx* mice initially but exacerbated it later on. Serum creatine kinase (CK) levels were significantly lower in 6- to 12-week-old *mdx:Smpd3* DKO mice than in *mdx* mice. However, at 28 weeks, moderately or significantly higher serum CK levels were observed in *mdx:Smpd3* DKO mice than in *mdx* mice. Furthermore, at 12 weeks of age, there were significantly fewer Evans blue dye (EBD: degree marker of myofiber damage)-positive muscle fibers in the tibialis anterior (TA) muscle of *mdx:Smpd3* DKO mice than in that of *mdx* mice. However, at 20 weeks of age, the number of EBD-positive muscle fibers in the TA of some *mdx:Smpd3* DKO mice was higher than that in *mdx* mice.

It has been reported that *nSMase2/Smpd3* inhibition reduces inflammatory responses via nuclear factor (erythroid derived 2)-like 2 activation. This activation then induces antioxidant response element-controlled genes and decreases the expression of inflammatory genes, including vascular cell adhesion molecule 1, intercellular adhesion molecule 1, monocyte chemoattractant protein-1, IL-1β, IL-6, and TNF-α, by inhibiting NF-κB activation in macrophages and endothelial cells, ultimately leading to suppression of the recruitment of monocytes to the endothelium and macrophage M1 differentiation [152]. Furthermore, *nSMase2/Smpd3* acts upstream of mitogen-activated protein kinase and NF-κB signaling pathways of TNF-α mediated inflammatory responses in monocytic cells/macrophages [153]. 

Furthermore, the systemic administration of EVs, particularly myotube-derived EVs, induced phenotypic rescue and mitigated pathological progression in dystrophic mice. This was due to improved membrane integrity, accompanied by the inhibition of intracellular influx and calcium-dependent calpain activation, which prevents the degradation of the destabilized dystrophin-associated protein complex [154]. These results suggest that, early in life, *nSMase2/Smpd3* ablation may have beneficial effects, with respect to excess inflammatory responses and myofiber membrane degeneration in *mdx* mice but that it may have adverse effects later on. In addition, genetic ablation of *nSMase2/Smpd3* improves muscle performance in mice with dystrophic phenotypes. The grip strength test demonstrated that, at 12 weeks of age, the muscle strength of *mdx* mice was significantly lower than that of wild-type mice, yet the muscle strength of *mdx:Smpd3* DKO mice significantly recovered. In addition, in an inclined treadmill running test, *mdx:Smpd3* DKO mice ran significantly longer than *mdx* mice at 16 and 60 weeks of age.

In addition to muscle degeneration in DMD, the loss of dystrophin in the brain has often been associated with nonprogressive cognitive deficits, behavioral disabilities, and enhanced fearfulness [155,156,157]. Thus, to investigate anxiety, emotionality, and the adaptive stress response to a novel environment in *mdx:Smpd3* DKO mice, the hole-board test was performed [151]. The results exhibited a loss of the *Smpd3* gene modulates anxiety behavior and stress responses, as well as the recovery of brain-derived neurotrophic factor (BDNF) expression, through exosomal miRNA in the hippocampus [151]. These findings suggest that DMD is not only a muscular disease but also a circulatory RNA disease, through intracellular communication. Furthermore, the signaling pathways modulated by the *nSMase2/Smpd3* protein might be novel therapeutic targets for DMD via the regulation of the expression levels of exosomal miRNAs using this *nSMase2/Smpd3* protein.

## 6. Adeno-Associated Virus (AAV)-Encapsulating EVs

Several clinical trials for DMD, using AAV vectors with various promoters, have been performed and are currently undergoing practical application ([15,48,66,82,158,159,160,161,162,163,164,165,166,167,168,169,170,171], Table 1). However, the risk of immune response from large doses of the AAV vector by systemic single administration to DMD patients was unresolved [172,173]. Thus, several strategies have been reported to suppress the immune response related to AAV vector administration. The first is the use of immunosuppressants, which are transduced into long-term skeletal muscles in non-human primates [174]. Second, examination of congenital and acquired immune tolerance induction methods showed that AAV vector-mediated microdystrophin, which is part of the full-length dystrophin gene, improved canine DMD pathology by the induction of immune tolerance. Moreover, systemic injection of the AAV9 vector with MSCs, which is a novel solution for treating DMD, into CXMDj of severe dystrophic phenotypes improves gene transfer and dystrophic phenotypes in skeletal muscle and heart functions by immune modulation [81,175]. Finally, vectors can be developed to avoid immune system attacks [81]. 

In addition, MSCs have been shown to improve pulmonary, cardiac, and distal skeletal muscle functions by their ability to fuse with dystrophic muscle, anti-inflammatory activities, and trophic factors that augment the activity of endogenous repair cells [176,177]. Furthermore, systemic administration of continuous IL-10 expressing MSCs by AAV vector into the CXMDj model indicated improvement of running capacity and recovery of tetanic force [178]. These findings showed the possibility of reducing the AAV administration dose in DMD patients to approximately 1/100 dose, with multiple administrations. However, some problems associated with gene therapy using AAV vectors remain. First, the systemic administration of AAV vectors carries a risk of carcinogenicity, hepatotoxicity, neurotoxicity, and thrombotic microangiopathy [172,173]. In addition, modification of the AAV capsid for construction of novel AAV vectors has unpredictable immunotoxicity and can damage non-target organs or exhibit unknown cross-reactivity of neutralizing antibody reaction for the AAV capsid in humans, with possible decreased productivity of the AAV vectors [179,180,181,182]. In addition to gene therapy, which is a treatment method for supplying or adding the genome editing strategy, which repairs mutations in abnormal genes or causes a loss of function of specific genes, has been explored [183,184,185]. Although the low efficiency of modifying target genes for the development of new in vivo technologies has been overcome, undesired effects have also increased, including the following: (1) off-target effects, (2) side effects with abnormal chromosome cleavege, and (3) carcinogenicity with suppression effects of p53 by homologous recombination [61,186,187,188]. In fact, clinical trials using genome editing have been discontinued because of suspicion of the development of chromosomal abnormalities. Therefore, as one of the novel gene therapy strategies, a drug delivery system for the AAV-encapsulating EV, which contains the AAV vector within the EV, was reported to be useful in the intervention of various target tissues in some diseases [189,190,191,192,193,194,195,196,197,198,199,200]. Furthermore, the effectiveness of gene therapy with AAV vectors is greatly affected by neutralizing antibodies [81,201,202,203]. Therefore, the induction of immune tolerance to AAV vectors has become an important issue for efficient and safe gene transduction. Nevertheless, it is possible to induce a reduction in immunogenicity to the AAV vector by encapsulating the AAV vector within EVs, which is naturally present in the body (as a natural transporter); it is expected that AAV vector transduction is more efficient than AAV vector administration alone [189,194,204]. In addition, AAV vectors administrated in vivo are consumed by innate immunity plasmacytoid dendritic cells, as mediated by Toll-like receptor 9 [205]. Therefore, attempts have been made to suppress these immune responses [81,174,177]. Furthermore, AAV-encapsulating EVs can also be introduced into in vivo barrier tissues such as the brain, eyes, and inner ear [189,190,191,192,193,194,195,196,197,198,199,200]. Therefore, repeated administration of AAV vectors would be possible because of dose reduction for in vivo administration with immunosuppression and/or EVs.

Although certain AAV serotypes, such as AAV9, efficiently, target muscle fibers, transduction of SCs has less reported. However, it was recently suggested that gene editing to the *dys* reading frame in the *mdx* mice was induced by transduction of AAV vector into the SCs [206,207]. Also, it was known that AAV vectors show their low immunogenicity, leading to persistence and long-term transgene expression of the vector, due to the reduced ability of AAV vectors to activate antigen-presenting cells [208]. However, neo-antigens, including viral capsid and the transgene product introduced by AAV in *mdx* muscle vectors elicit cellular and humoral immune responses, which can be reduced using immune modulatory treatments targeting T cells, B cells, and pro-inflammatory processes [208,209]. It was indicated that treatment with the immune modulatory drugs, including cytotoxic T-lymphocyte-associated protein 4 (CTLA4)-linked IgG2a, Rituximab, Prednisolone, and VBP6 enhanced the beneficial effects of AAV-microdystrophin therapy for force generation [208]. 

Also, interaction between the SCs and EVs was critical event for myogenesis and angiogenesis during skeletal muscle regeneration [120,210]. In addition with this reports, EVs from muscular progenitor cells (MPCs) treated with a histone deacetylase inhibitor (HDACi), which has been shown to increase muscle regeneration in *mdx* mice, were enriched myoangiogenesis-related miRNAs, such as miR-181a, miR-17, miR-210, miR-107, miR-19b, Let-7e-5p, miR-26a, and miR-103 compared with EVs from untreated MPCs [211]. Further, EVs released by MSCs such as fibro-adipogenic progenitors (FAPs) transfer miRNAs, which cooperatively target therapeutic pathways related with regeneration, fibrosis, and inflammation, to SCs [211]. Since increase of miR-206 level within the EVs released by FAPs in muscles indicated the essential roles of the miR-206 for the EV-induced expansion of MPCs and regeneration of dystrophic muscles [211]. These reports suggest that miRNA transduction via the EVs into MPCs will be novel therapeutic intervention in muscular dystrophy. 

## 7. Conclusions

Currently, some therapeutic strategies for DMD have been clinically applied, but no curative treatment for DMD has been yet established. Although AAV vectors are considered to be a promising therapeutic strategy for treating DMD, because of their safety and usefulness, their use still has concerns, such as adverse effects caused by a single high-dose administration. However, it was clarified that the induction of immune tolerance by MSCs improved the therapeutic effects by alleviating the action of immune cells on AAV vectors. Furthermore, AAV-encapsulated EVs could be a novel therapeutic strategy for DMD because they can escape from neutralizing antibodies.

## Figures and Tables

**Figure 1 ijms-23-01551-f001:**
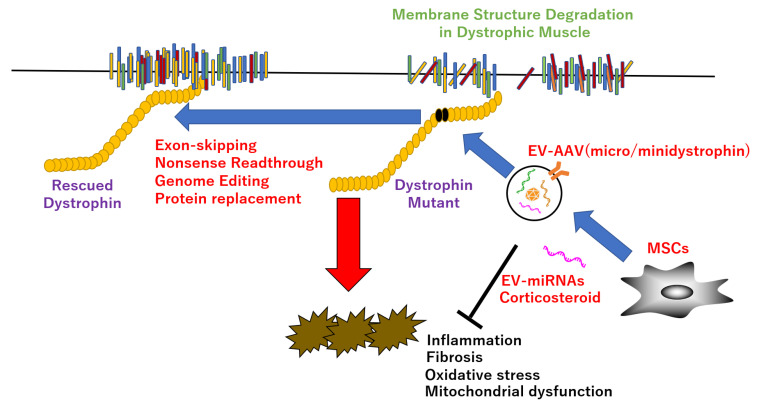
Therapeutic strategy in DMD. EV-encapsulated AAV or miRNAs improve dystrophic pathology, such as inflammation, fibrosis, oxidative stress, and mitochondrial dysfunction, etc., via direct rescue of dystrophin mutant by exon-skipping, nonsense readthrough, genome editing, protein replacement, or indirect recovery of dystrophin functions by improvement of the downstream targets.

**Table 1 ijms-23-01551-t001:** AAV vector serotypes used in Clinical trials for DMD.

Serotypes	Transgene	Promoter	Phase	Company	Dose/Route	Clinical Trials ID	Annotation
AAV2.5	Mini-Dys	CMV	I	AsklepiosBiopharmaceutical, Nationeide Chilren's Hospital	2 cohorts	NCT00428935	
AAV9	Mini-Dys	humanmuscle specific	Ib	Pfizer, PF-06939926	2 cohorts, single i.v. inj.	NCT03362502	
III	NCT04281485	Muscle weakness, myocarditis ⇒ hold
AAVrh74	µDys	MHCK7	I	Sarepta (SRP9001), Roche, Nationwide Children’s Hospital	2 × 10^14^ vg/kg, i.v.	NCT04626674	Stable dose of corticosteroids throughout trial
II	NCT03769116
I/II	NCT03375164
III	NCT05096221
AAV9	µDys	CK8	I/II	Solid Biosci., SGT-001	2 dose, single i.v.	NCT03368742	Once hold, protocol changed(C5/C1 inhibitor use)
AAV9	U7-snRNA (ACCA)		I/IIa	Audentes Therapeutics, Nationwide Children’s Hospital	Cohort 1, minimal effective dose, peripheral limb vein inj.	NCT04240314	
AAV3b	U7-snRNA-E53		I/II	Genethon, Institute of Myology		NCT01385917	AAV-mediated Exon53 skipping
AAV8	µDys	spC5-12	I/II/III	Genethon, Sarepta

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
