# Peer review of "Therapeutic Application of Extracellular Vesicles-Capsulated Adeno-Associated Virus Vector via nSMase2/Smpd3, Satellite, and Immune Cells in Duchenne Muscular Dystrophy"

_ijms, 2022, doi:10.3390/ijms23031551_

Round 1
Reviewer 1 Report
The authors of this review showed possible approaches to the treatment of Duchenne muscular dystrophy, a severe, almost incurable disease of the muscular system. This is a fairly traditional work, and I have several recommendations that, in my opinion, will improve its presentation:
- Authors need to show the aim of this review at the end of the abstract and introduction. It is not entirely obvious now.
- In the second section, the authors should separate approaches to the treatment of the actual cause of DMD - dystrophin deficiency (dystrophin-targeted therapies) and approaches to the therapy of downstream pathological changes. In the first case, in addition to the approaches indicated on lines 92-96, the authors should mention protein replacement therapies (primarily utrophin) (doi: 10.1096/fj.201800081R). In the second case, the authors should mention an improvement in calcium homeostasis, a decrease in oxidative stress in muscle tissue (doi: 10.1002/mus.22047; doi: 10.1016/j.nmd.2019.10.008; doi: 10.3390/ijms22189780), and an improvement in mitochondrial function and biogenesis (doi: 10.3390/ijms21228763; doi: 10.1038/nm1736; doi: 10.1016/j.phrs.2021.105421). In this case, treatment with corticosteroids (specific names of drugs should also be mentioned) should also be attributed more likely to the therapy of concomitant destructive processes, namely inflammation. Perhaps the authors will find additional ways of correcting secondary changes that deserve mention.
- The authors should also accompany the review with a qualitative illustration showing possible targets for dystrophin deficiency therapy and the indicated downstream pathological changes. This will greatly improve the presentation of the work.
- Lines 77-79. It is necessary to clarify what specific proteins the authors are talking about.
Author Response
Thank you very much for your reviewing. According to the reviewers’ comments and suggestions, we revised our manuscript by point-by-point response. We wish to express our appreciation to the comments and suggestions, which helped to improve the manuscript.
1. Authors need to show the aim of this review at the end of the abstract and introduction. It is not entirely obvious now.
In accordance with the reviewers’ comments, we added the sentences about aim of this review on line 29 -30 in page 1, and on line 96 -97 in page 3 indicated by yellows.
2. In the second section, the authors should separate approaches to the treatment of the actual cause of DMD - dystrophin deficiency (dystrophin-targeted therapies) and approaches to the therapy of downstream pathological changes. In the first case, in addition to the approaches indicated on lines 92-96, the authors should mention protein replacement therapies (primarily utrophin) (doi: 10.1096/fj.201800081R). In the second case, the authors should mention an improvement in calcium homeostasis, a decrease in oxidative stress in muscle tissue (doi: 10.1002/mus.22047; doi: 10.1016/j.nmd.2019.10.008; doi: 10.3390/ijms22189780), and an improvement in mitochondrial function and biogenesis (doi: 10.3390/ijms21228763; doi: 10.1038/nm1736; doi: 10.1016/j.phrs.2021.105421). In this case, treatment with corticosteroids (specific names of drugs should also be mentioned) should also be attributed more likely to the therapy of concomitant destructive processes, namely inflammation. Perhaps the authors will find additional ways of correcting secondary changes that deserve mention.
In accordance with the reviewers’ comments, we corrected this section on line 100 - 111 in page 3 indicated by yellows.
3. The authors should also accompany the review with a qualitative illustration showing possible targets for dystrophin deficiency therapy and the indicated downstream pathological changes. This will greatly improve the presentation of the work.
In accordance with the reviewers’ comments, we added the Figure1 in page 4 indicated by yellows.
4. Lines 77-79. It is necessary to clarify what specific proteins the authors are talking about.
In accordance with the reviewers’ comments, we added the sentences about the protein of sarcolemmal tears on line 80 - 83 in page 3 indicated by yellows.
Reviewer 2 Report
In the present manuscript, the authors systematically reviewed therapeutic strategies in Duchenne Muscular Dystrophy (DMD). The authors firstly introduced characteristics of DMD, followed by description of different therapy strategies. The authors also talked about impact of satellite and immune cells, the role of non-invasive biomarkers, nSMase2/SMPD3 in DMD therapy.
This review was well-thought but not well-organized and well-executed. There are several concerns that should be addressed before accepting this manuscript for publication:
- The title cannot fully reflect the content of the main text.
- The structure of the article is logically unclear. For example, the authors talked about the role of satellites and immune cells through a lot of description, but did not show the important relationship with AAV or EVs. Other parts of the main text also showed the same problem.
- Please discuss how miRNA can be applied into DMD therapy through AAV or EV.
- Too many typos and inconsistent formats in the main text. For example, line 103. “Additionally,” not “Additinally”; line 111 “Dysfunctions” not “Dusfunctions”; line 271 “nSMase2/SMPD3” or line 272/276 “nSMase2/Smpd3” etc.
Author Response
Thank you very much for your reviewing. According to the reviewers’ comments and suggestions, we revised our manuscript by point-by-point response. We wish to express our appreciation to the comments and suggestions, which helped to improve the manuscript.
1. The title cannot fully reflect the content of the main text.
In accordance with the reviewers’ comments, we corrected title of this manuscript on line 1 – 4 in page 1 indicated by yellows.
2. The structure of the article is logically unclear. For example, the authors talked about the role of satellites and immune cells through a lot of description, but did not show the important relationship with AAV or EVs. Other parts of the main text also showed the same problem.
In accordance with the reviewers’ comments, we added the sentences about the relationship of satellite and immune cells with AAVs and EVs on line 419 - 439 in page 13 to 14 indicated by yellows.
3. Please discuss how miRNA can be applied into DMD therapy through AAV or EV.
In accordance with the reviewers’ comments, we added the sentences about the miRNA application for DMD on line 433 - 446 in page 13 to 14 indicated by yellows.
4. Too many typos and inconsistent formats in the main text. For example, line 103. “Additionally,” not “Additinally”; line 111 “Dysfunctions” not “Dusfunctions”; line 271 “nSMase2/SMPD3” or line 272/276 “nSMase2/Smpd3” etc.
In accordance with the reviewers’ comments, we corrected some typos and inconsistent formats indicated by yellows.
Round 2
Reviewer 1 Report
The manuscript is appropriately revised, and it can be published in the current state.
Reviewer 2 Report
The authors revised the manuscript extensively according to the reviewer's comments, therefore it is acceptable to publish this manuscript in IJMS.